# A Salinity–Temperature Sensor Based on Microwave Resonance Reflection

**DOI:** 10.3390/s22155915

**Published:** 2022-08-08

**Authors:** Darek J. Bogucki, Tom Snowdon, Jennifer C. Doerr, Joseph E. Serafy

**Affiliations:** 1Department of Physical and Environmental Sciences, Texas A&M University-Corpus Christi, Corpus Christi, TX 78412, USA; 2Independent Researcher, Miami, FL 33133, USA; 3Southeast Fisheries Science Center, Galveston, TX 77551, USA; 4Southeast Fisheries Science Center, Miami, FL 33149, USA

**Keywords:** aquatic salinity measurements, aquatic temperature measurements, environmental monitoring

## Abstract

We developed and tested a microwave in situ salinity sensor (MiSSo) to simultaneously measure salinity and temperature within the same water sample over broad ranges of salinity (S) (3–50 psu) and temperature (T) (3–30 °C). Modern aquatic S sensors rely on measurements of conductivity (C) between a set of electrodes contained within a small volume of water. To determine water salt content or S, conductivity, or C, measurements must be augmented with concurrent T measurements from the same water volume. In practice, modern S sensors do not sample C and T within the same volume, resulting in the S determination characterized by measurement artifacts. These artifacts render processing vast amounts of available C and T data to derive S time-consuming and generally preclude automated processing. Our MiSSo approach eliminates the need for an additional T sensor, as it permits us to concurrently determine the sample S and T within the same water volume. Laboratory trials demonstrated the MiSSo accuracy of S and T measurements to be <0.1 psu and <0.1 °C, respectively, when using microwave reflections at 11 distinct frequencies. Each measurement took 0.1 μs. Our results demonstrate a new physical method that permits the accurate S and T determination within the same water volume.

## 1. Introduction

Accurate and precise salinity and temperature records are fundamental measurements in estuarine, coastal, and ocean monitoring programs [1,2,3]. For the most part, these programs rely on networks of stationary logging instruments with separate conductivity and temperature (C and T) sensors. Conventionally, salinity is estimated from conductivity and temperature measurements. Conductivity is usually determined by applying an electrical current between two electrodes, and measuring voltage change and temperature via a temperature-sensitive resistor. Monitoring networks can be expensive to maintain given the harsh physicochemical conditions in which sensing instruments are deployed, and experience problems associated with biofouling [4]. The same is true for sensors mounted on moving robotic platforms such as Argo floats [5], gliders [6], and autonomous underwater vehicles, which are increasingly considered to be a more cost-effective approach to environmental data collection than recording from crewed vessels. These platforms produce a large amount of temperature and conductivity data that necessitate an automated correction step, whether stationary or moving.

That need for two different sensors, C and T, and their spatial separation result in several undesirable effects that need to be corrected for when carrying out high-accuracy salinity and water density measurements [7,8]. The first is the correction for response time, as each C or T sensor has a different measurement response times while being physically separated by some distance. To mitigate the time lag between C and T when calculating S, the two independent measurements need to be temporally aligned, such that each CT record represents measurements for the same parcel of water. This time shift should account for the instrument’s sample rate and, if mounted on a moving platform or pumped, for the water flow rate [8]. A second adjustment is a thermal mass correction needed to account for the thermal mass of the conductivity cell and its effect on the resulting salinity calculation. However, the temperature is measured outside the conductivity cell, while conductivity is measured inside the cell. In addition, the conductivity cell itself can store heat from the surrounding water inside the wall of the cell, resulting in the heating or cooling of new water parcels as they pass through the cell. As a result of this thermal lag, the paired conductivity and temperature used to calculate salinity may result in erroneous salinity values, especially when measuring across temperature gradients [7,8]. That thermal lag correction [7] can result in measured salinity varying by 0.1 psu (for the SBE-41), thus limiting salinity sensor accuracy. The third correction is related to the different response times of C and T sensors. Typically, a C sensor has a response time of approximately 0.001 s, while a T sensor has a response time [9] of >0.1 s. A T sensor thus constrains the salinity measurement response time to 0.1 s, rendering it suboptimal for measuring salinity gradients from fast-moving platforms. To obtain S measurements when the response time of <0.1 s is needed, a complicated spectral correction for T response is required [9].

The present work applies the novel approach of resonant microwave reflectometry to derive water salinity and temperature from microwave reflection measurements within the same water sample. The microwave in situ salinity sensor (MiSSo), specifically its antenna, generates an electromagnetic (EM) field within the water parcel surrounding the antenna. The MiSSo sensing volume of cylindrical geometry is determined by the antenna length and the wavelength of EM radiation. The resonant microwave reflectometry approach yields T and S, which are measured within the same sample volume and at the same time without the need for separate temperature measurements and other corrections. The microwave resonance reflection approach differs from conventional conductivity or resistance sensing in that it: (a) eliminates thermal and temporal lag effects, rendering it appropriate for rapidly moving platforms such as AUVs, UAVs, and water surface drones; and (b) reduces maintenance costs associated with biofouling because its sensing elements are smooth wipeable surfaces that are resistant to the gradual accumulation of organisms (e.g., algae, bacteria, invertebrates). The work presented here describes results from controlled laboratory trials where MiSSo observations of salinity were compared with those generated by a conventional instrument relying on CT data to determine S.

## 2. Materials and Methods

### 2.1. Microwave Resonance Reflectometry

A microwave reflectometry sensing system, as shown in Figure 1A, including MiSSo, consists of a signal source (TX), a microwave antenna electromagnetically (EM) interacting with the water sample, and a receiver (RX) that combines the reflected signal from the probe and a reference signal from the source, thus generating an interferometric output signal. From a physical standpoint, the MiSSo senses the sample’s relative dielectric permittivity, $\epsilon_{r}$, within the antenna sensing volume. Dielectric permittivity is the ability of a substance to hold an electrical charge and its value is frequency-dependent. Here, $\epsilon_{r}=\epsilon_{r}^{\prime }+i\epsilon_{r}^{\prime \prime }$, where $\epsilon_{r}^{\prime }$ is the real- and $\epsilon_{r}^{\prime \prime }$ the complex-valued part of the $\epsilon_{r}$.

### 2.2. MiSSo Operating Principle

The MiSSo sensor utilizes a microwave resonance reflectometry approach to obtain sample values of temperature and salinity from sample reflection spectra. The reflection spectra of the sample are obtained by sweeping the frequency of source TX and collecting the signal amplitude by the RX as $S_{11}\left(f\right)$ data. At a given frequency, f, the reflected signal from the probe, $S_{11}\left(f\right)$, carries information regarding the sample’s permittivity, Figure 1A,B. The $S_{11}\left(f\right)$ function is referred to as the reflection spectrum. Physically, the $S_{11}\left(f\right)$ spectrum is related to the measured voltage of the standing wave created by interference of the TX signal with its reflection off the antenna. The important feature of the microwave resonant reflection spectrum is the presence of a resonance or a minimum of spectrum $S_{11}\left(f\right)$ located at the frequency, $f_{0}$, such that $S_{11}\left(f_{0}\right)$ attains a minimal value there. Sample parameters such as the salinity and temperature of the sample can then be extracted from $S_{11}\left(f\right)$ using a suitable mathematical model such as the one presented in [10] and reproduced here as Equations (Equation 1)–(Equation 3).

The reference water sample reflection spectra, $S_{11}^{0}\left(S_{0},T_{0},f\right)$, are taken here as measured at the lowest bounds of analyzed (*S*, *T*), namely, at $S_{0}=3$ psu and $T_{0}=10$ °C. In response to a change in seawater salinity or temperature, the reflection spectrum shifts its location from $f_{0}$ to $f_{0}+\Delta f$. The new reflection spectrum minimum, located now at frequency $f_{0}+\Delta f$, can be found [10] as follows:(1)$$\Delta f=\frac{1}{2}f_{0}V_{c}\frac{\epsilon_{r}^{\prime }{}^{\left(h\right)}-\epsilon_{r}^{\prime }{}^{\left(c\right)}}{A+\epsilon_{r}^{\prime }{}^{\left(h\right)}}$$
where: $f_{0}$ is the frequency where the $S_{11}\left(f\right)$ attains its local minimal value, $S_{11}\left(f_{0}\right)$, the $\epsilon_{r}^{\prime }{}^{\left(h\right)}$ is the dielectric constant of the host (reference water sample), $\epsilon_{r}^{\prime }{}^{\left(c\right)}$ is the dielectric constant of the dissolved material (salt), and $V_{c}$ is the volumetric concentration (per unit volume) of dissolved salt when compared to the reference sample. *A* is a constant that depends on the sensor geometry [10]. The new reflection spectrum $S_{11}\left(S,T,f\right)$, with its minimum now located at $f_{0}+\Delta f$, can now be related [10] to the reference reflection spectrum as follows:(2)$$\Delta S_{11}\left(S,T,f\right)\equiv S_{11}\left(S,T,f\right)-S_{11}^{0}\left(S_{0},T_{0},f\right).$$
where $\Delta S_{11}\left(S,T,f\right)$ is the deviation of the reflection spectra from its reference set of values given by $S_{11}^{0}\left(S_{0},T_{0},f\right)$.

To understand which water properties affect the value of $\Delta S_{11}\left(S,T,f\right)$ for small frequency change Δf and in vicinity of reflection minimum $f_{0}$, the term $\Delta S_{11}\left(S,T,f\right)$ can be expressed [10] as:(3)$$∥\Delta S_{11}\left(S,T,f\right)∥=2\;∥S_{11\;min}∥{\left(Q\frac{\Delta f}{f_{0}}\right)}^{2}$$
where $S_{11\;min}$ denotes $S_{11\;min}=S_{11}\left(S,T,f_{0}\right)$, and *Q* is the resonance quality factor. The model, as shown in Equations (Equation 1)–(Equation 3), demonstrates that the reflection spectrum response to seawater is proportional to small salinity or temperature changes, and depends on the value of the dielectric permittivity contrast between an analyzed sample and the reference.

In principle, the reflection spectrum, $S_{11}\left(f\right)$, is sensitive to other seawater contaminants and their concentration [10,11,12], but the work presented here is focused on the temperature and the salinity contributions.

The seawater dielectric permittivity $\epsilon_{r}^{\prime }{}^{\left(c\right)}$ depends on *S* and *T* [13], which results in $\Delta S_{11}\left(f_{0}\right)$ becoming a function of salinity, temperature, and frequency, i.e., $\Delta S_{11}\left(S,T,f_{i}\right)$, where $f_{i}$ is a selected frequency. To determine *S* and *T* from $\Delta S_{11}\left(S,T,f_{i}\right)$ measurements, we selected several frequencies $f_{i}$ in the $S_{11}\left(f_{i}\right)$ spectra that were used in *T* and *S* determination; Figure 1C. In our experiments, we tested a varying number *i* of applied frequencies, $f_{i}$, i= 2, 4, 6, 8, and 11, to determine function $\Delta S_{11}\left(S,T,f_{i}\right)$. For each of the selected frequencies $f_{i}$, $\Delta S_{11}\left(S,T,f_{i}\right)$ was determined via the least-squares fit of a polynomial model of $\Delta S_{11}\left(S,T,f_{i}\right)$ to the set of $S_{11}\left(f_{i}\right)$ for known *S* and *T* values. *T* and *S* values were obtained by concurrent measurements using a conventional CT sensor (MiniSonde 4a, Hydrolab). We refer to the modeled function, measured for a known sample set values of (*S*, *T*), as the calibration function, $\Delta S_{11cal}\left(S,T,f_{i}\right)$. Once model function $\Delta S_{11cal}\left(S,T,f_{i}\right)$ had been established, it was used to find *S* and *T* from measured $S_{11}\left(f\right)$.

The steps to obtain samples *T* and *S* were then as follows: (1) measure sample $S_{11}\left(f\right)$; (2) subtract $S_{11}\left(f\right)$ from the reference value $S_{11}^{0}\left(S_{0},T_{0},f\right)$ to obtain sample $\Delta S_{11}\left(S,T,f\right)$; and (3) use sample $\Delta S_{11}\left(S,T,f\right)$ to carry out a two-dimensional least-squares fit to the calibration function, $\Delta S_{11cal}\left(S,T,f_{i}\right)$, to retrieve samples *T* and *S*.

### 2.3. Experimental Setup

The implemented MiSSo operates in the 1–3 GHz microwave band, as shown in Figure 1A,B. The prototype MiSSo sensor version was equipped with three vertical monopole antennas, as shown in Figure 1B. The presence of three slightly different antennas allowed for us to evaluate the effect of antenna geometry on $S_{11}\left(f\right)$ reading accuracy. The laboratory setup consisted of a suite of instruments for the control and data acquisition of sample water temperature and salinity using a system of sensors, loggers, pumps, and a cooler. Instruments used in the experiments were:A commercial salinity and temperature sensor (MiniSonde 4a, Hydrolab) with a 1 Hz sampling rate and manufacturer-specified accuracy of 0.1 psu and 0.1 °C.Three FP07 (GE) fast thermistors connected to an Agilent 34970A data acquisition unit to sample water temperature with 5 Hz rate with 0.05 °C accuracy.The current MiSSo prototype equipped with three monopole antennas. Each antenna was approximately 1.7 cm long and was separated from the other antennas by either 5 or 2 cm distance. Each monopole antenna had resonance at approximately $f_{0}=2.65$ GHz.Vector network analyzer (VNA), R&S FPC 1500, (0–3 GHz range). The FPC 1500 acquired 2500 values of $S_{11}\left(f\right)$ within a 1–3 GHz range with few Hz and <0.05 dB $S_{11}$ accuracy over a few minutes. For stability, we collected three $S_{11}\left(f\right)$ spectra for each *S* and *T* value.A computer-controlled microwave switch, 0–18 GHz, (Pasternack).A set of 1 m long coaxial cables (Pasternack) connecting antennas to the VNA.A magnetic water stirrer to ensure water samples were homogeneous during measurements.A computer-controlled Peltier water cooler and computer-controlled water pump.

### 2.4. Data Management and Control

Overall data management and control were achieved using a Dell laptop connected via the network to each instrument. Instrument control was implemented using software coded in MATLAB. The current prototype MiSSo calibration and test data set spanned 3–50 psu and 12–30 °C ranges. Each set of $S_{11}\left(f\right)$ values were acquired over time intervals ranging from 2 to 15 min. Within these times, we collected over 100 $S_{11}$ frequency sweeps within the 1 to 3 GHz range with 2000 discrete frequency steps. The 100 sweeps within the 1 to 3 GHz range were then averaged to a single $S_{11}\left(f\right)$ value over the frequency range, each consisting of 2000 frequency measurements, $S_{11}\left(f_{i}\right)$. During each measurement, with the aid of a CT sensor (MiniSonde 4a, Hydrolab), the standard deviation (δ) of sample *T* or *S* was determined. It was found that the deviations were δT=0.1 °C and δS=0.25 psu, respectively.

### 2.5. Components of the MiSSo Measured Signal

The MiSSo raw signal $S_{11}\left(f\right)$ was a sum of two signals:Signal (1):the signal reflected off the sample;Signal (2):the standing wave signal due to the TX and RX impedance mismatch.

The MiSSo antennas were characterized by the main resonance at $f_{0}$ = 2.65 GHz corresponding to the 1.7 cm long monopole antenna, as shown in Figure 1C. The characteristic rapid oscillations in the $S_{11}\left(f\right)$ of the Signal (2) were the result of the microwave reflection within long cables connecting the microwave network analyzer, the microwave switch, and the antenna. Signal (2) was independent of Signal (1). In Figure 1C, reflected Signal (1), $S_{11}\left(f\right)$, is marked red, while the black line represents the sum of Signals (1) and (2) as measured by MiSSo. The standing wave component, (2), can be filtered out from the measured $S_{11}\left(f\right)$ signal. The presence of Signal (2) did not obscure the *T* and *S* retrievals. For the analysis of seawater composition, we then used raw $S_{11}\left(f\right)$ data, i.e., consisting of the sum of (1) and (2), as shown in Figure 1C.

## 3. Results and Discussion

The calibration dataset needed to create model function $\Delta S_{11cal}\left(S,T,f_{i}\right)$ was obtained over a range of different frequencies, salinities (3–50 psu), and temperatures (12–30 °C), consisting of over 6 million $S_{11}$ measurements. We illustrate the process of the salinity determination from MiSSo data in Figure 2A, where we used model function $\Delta S_{11cal}\left(S,T,f_{i}\right)$ obtained at the four selected (*i* = 1–4) frequencies, $f_{i}$: 2.1222, 2.3411, 2.7454, 2.9255 GHz and over a range of salinities (*S* = 3–50 psu) and temperatures (*T* = 12–30 °C). An example of the $S_{MiSSo}$ and $T_{MiSSo}$ retrieval from the MiSSo measured $S_{11}\left(f\right)$ is illustrated in Figure 2A. Here we present a color-coded map of $S_{11}\left(f_{i}\right)$ deviation from the model function $\Delta S_{11cal}\left(S,T,f_{i}\right)$. The color bar represents the $\log_{10}$ of the absolute deviation between the sample value and the modeled $\Delta S_{11cal}\left(S,T,f_{i}\right)$. That is, the −2 (deep blue) value of that difference corresponds to a 0.01 deviation from the model function. The smallest difference on that map is characterized by coordinates $S_{MiSSo}$, $T_{MiSSo}$ represents the point within the (*T*, *S*) space corresponding to the MiSSo-derived temperature and salinity of the sample, $T_{MiSSo}$ and $S_{MiSSo}$. In Figure 2A, deep blue is the smallest deviation value corresponding to MiSSo retrieved values of $T_{MiSSo}=15.02$ °C and $S_{MiSSo}=10.05$ psu. Sample *T* and *S* values were independently measured by the CT sensor (MiniSonde 4a), and were $T_{MiniSonde}=15.06$ °C and $S_{MiniSonde}=11.00$ psu. However, these $S_{MiniSonde}$ and $T_{MiniSonde}$ values are somewhat unrealistic, as the laboratory temperature measurement accuracy using the Hydrolab’s MiniSonde 4a and fast thermistors were only 0.1 °C and 0.25 psu, respectively (1 × standard deviation).

To establish the accuracy of the MiSSo *T* and *S* retrievals, we carried out a comparison between the CT sensor (MiniSonde 4a), and MiSSo-measured salinity and temperature over 10,000 *T* and *S* samples within the range of measured salinities (3–50 psu) and temperatures (12–30 °C). For the comparison to be more realistic of field conditions, we degraded the MiSSo-measured $S_{11}\left(f\right)$ by adding random noise of 0.05 dB to each $S_{11}\left(f_{i}\right)$ and then carried out retrievals as shown in Figure 2A. This external EM noise to the antenna simulates the effect of antenna biofouling, the presence of particles within the sample, and/or external EM noise. The standard deviation for the four frequencies of that degraded signal is presented as a normalized probability density function of differences ($T_{MiSSo}-T_{MiniSonde}$) and ($S_{MiSSo}-S_{MiniSonde}$), Figure 2B. For the four used frequencies, 2.1222, 2.3411, 2.7454, 2.9255 GHz, the standard deviation between MiSSo and MiniSonde 4a retrieved *T* and *S* was $\delta (S_{MiSSo}-S_{MiniSonde})=0.09$ psu and $\delta (T_{MiSSo}-T_{MiniSonde})=0.06$ °C, where δ is one standard deviation, Figure 2B. In the vicinity of the resonance $f_{0}$, the MiSSo-measured reflection spectrum $S_{11}\left(f\right)$ permits the derivation of the MiSSo-measured *S* and *T*, i.e., $S_{MiSSo}$ and $T_{MiSSo}$ values, when $S_{11}\left(f_{i}\right)$ is known for at least two distinct frequencies, $f_{1}$ and $f_{2}$. This strategy of ($S_{MiSSo}$, $T_{MiSSo}$) retrieval is effective provided the information in each pair consisting of $S_{11}\left(f_{1}\right)$ and $S_{11}\left(f_{2}\right)$ is independent. We additionally verified the hypothesis that an increase in the number of independent frequencies (up to i=11) increases the *T* and *S* retrieval accuracy for a given measurement. The accuracy of MiSSo measured salinity and temperature as a function of several used frequencies and added noise (either 0.05 dB or 0.1 dB) is shown in Table 1. There, for example, when using 11 frequencies and with noise of approximately 0.05 dB, the accuracy of the *S* and *T* retrievals was δS=0.11 psu and δT=0.07 °C.

An example of a time series of MiSSo-measured *T* and *S* values is presented in Figure 3A–C. MiSSo data are plotted with concurrent CT sensor (MiniSonde 4a)-measured *T* and *S*. The effect of external noise, such as associated with biofouling, can be observed in Figure 3B,C where the addition of 0.05 dB noise to the MiSSo reading increased uncertainty in the MiSSo retrieved salinity readings.

## 4. Conclusions

Microwave resonance reflection appears to be a very promising alternative to the conventional approach of measuring salinity and temperature using conductivity electrodes and temperature-sensitive resistors. Our laboratory trials demonstrated MiSSo accuracy of at least 0.1 psu and 0.1 °C per single multifrequency measurement with 0.05 dB background noise, as shown in Table 1.

To estimate the level of accuracy that MiSSo could attain when carrying out multiple measurements of the same water sample, we estimated the following. Oceanic water and the water in our experiment was a turbulent flow that was characterized by chaotic changes in *T* and *S*. That turbulent flow, at small scales over a few cm, was characterized by temporal variability ranging from 0.2 to 10 s [14,15]. This means that the water sample under measurement was characterized by approximately invariant *T* and *S* when the measurement was carried out over a time interval shorter than <0.2 s [9]. For a turbulent flow, if we carried out an *N* measurement over a time interval shorter than 0.2 s, their average would yield the unbiased estimator of the true mean, while the standard deviation of the error [16] would be $\delta T/\sqrt{N}$ or $\delta S/\sqrt{N}$, where the δT and the δS are the population variances (Table 1).

In principle, the MiSSo can carry out 100 consecutive measurements within 0.01 s, a time interval which is much less than the 0.2 s characterizing the temporal variability of the measured sample. For the 100 consecutive MiSSo measurements obtained over 0.01 s, MiSSo accuracy could result in *T* and *S* standard deviations of 0.01 psu and 0.01 °C. The above analysis is approximate, as a more accurate approach would involve measurements of *S* and *T* spatial and temporal spectra and performing a detailed spectral analysis [16]. We plan to conduct that analysis in future experiments.

MiSSo accuracy and speed exceeded those of most commercial CT sensors, as the typical thermistor response time used in modern CT sensors is over 0.1 s [7], and Table 2.

Our work demonstrates that seawater salinity and temperature can be simultaneously measured with a single sensor (antenna).

MiSSo eliminates thermal and temporal lag effects associated with conventional salinity measurements permitting automated data processing of large datasets without the need to remove artifacts associated with available commercial sensors, where care must be taken to account for separate T and C sensors. MiSSo can also be implemented with a flat wipeable configuration [10]. In that wipeable configuration, MiSSo lowers maintenance costs associated with biofouling, as it enables long-duration deployments. In the case of moored oceanic salinity sensors, maintenance costs can be significant, as in biologically productive waters, biofouling forces instrument operators to periodically clean the CT sensor, thus incurring vessel-fuel and other costs that can easily exceed thousands of dollars a day [17].

A comparison of MiSSo performance with representative CTDs is presented in Table 2.

## Figures and Tables

**Figure 1 sensors-22-05915-f001:**
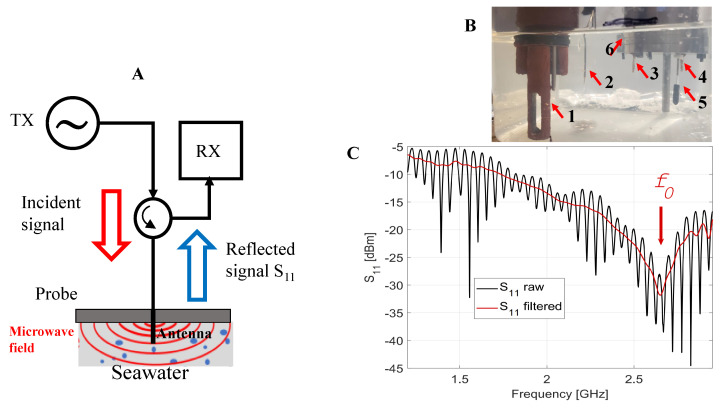
(**A**) Schematics of the microwave reflection measurement system: TX, transmitter; RX, receiver and the probe containing the antenna. (**B**) Experimental setup and sensor location: (1) MiniSonde; (2) fast thermistors FP07; (3–5) 1/2 wavelength monopole antennas; (6) submerged part of the sensor. (**C**) Example of sample reflection measurement $S_{11}\left(f\right)$ with S=10 psu and T=20 °C as a function of frequency *f*. The measured $S_{11}\left(f\right)$ (black line) can be decomposed into the Signal (1) reflected off the sample (red line) and the signal caused by TX/RX impedance mismatch (not shown here). $f_{0}$ is the main resonance frequency.

**Figure 2 sensors-22-05915-f002:**
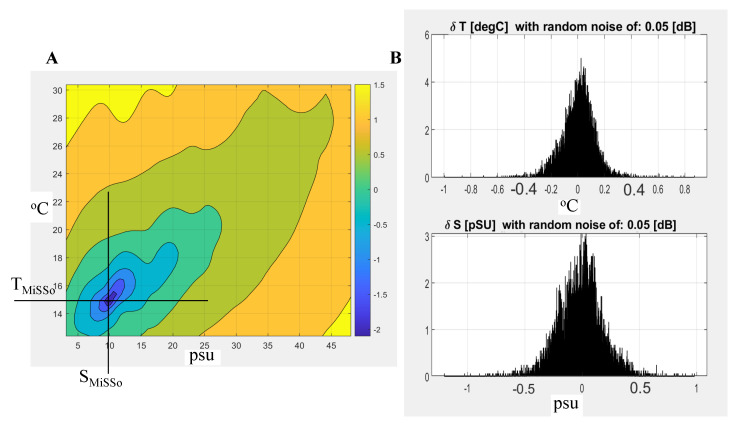
(**A**) An example of temperature and salinity determination for four frequencies. The color-coded least-squares deviation of a point located at ($S_{MiSSo}$, $T_{MiSSo}$) from minimum value of $\Delta S_{11}\left(S,T,f_{i}\right)$, for four selected $f_{i}$ frequencies 2.1222, 2.3411, 2.7454, 2.9255 GHz. In this example, the MiSSo-found *S* and *T* was $S_{MiSSo}=10.05$ psu and $T_{MiSSo}=15.02$ °C. Concurrently, sample *T* and *S* values were independently measured by the CT sensor (MiniSonde 4a) and were $T_{MiniSonde}=15.06$ °C and $S_{MiniSonde}=11.00$ psu. (**B**) The histogram of deviations $\delta (S_{MiSSo}-S_{MiniSonde})$ and $\delta (T_{MiSSo}-T_{MiniSonde})$ of the measured *T* and *S* from the seawater *S* and *T* at selected four frequencies: 2.1222, 2.3411, 2.7454, 2.9255 GHz. Here, the 0.05 dB noise was added to the received signal.

**Figure 3 sensors-22-05915-f003:**
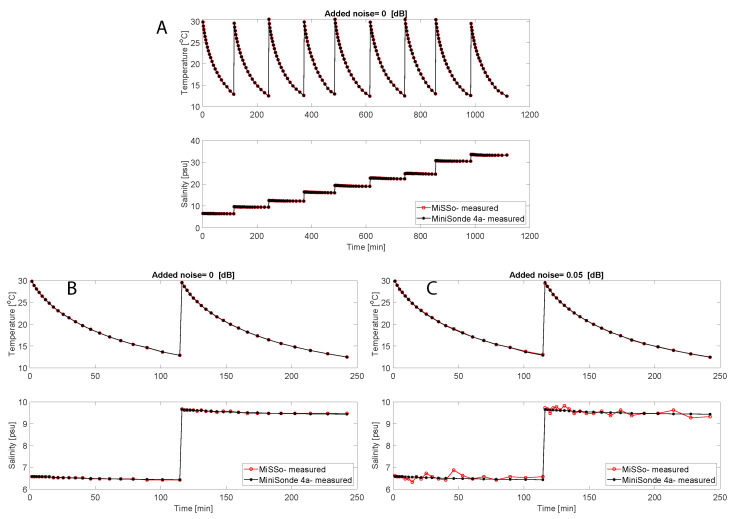
Time series of MiSSo-retrieved temperature and salinity compared to readings of commercial CTD sensor MiniSonde 4a. Red denotes the MiSSo data, and black the MiniSonde 4a measurements. In each experiment of ≈120 min duration, salinity was approximately constant, while the temperature was computer-controlled between 12 and 30 °C. The abrupt increases in salinity reflect the change in sample salinity between (120 min) experiments. (**A**) Time series of *T* and *S* data spanning 4 days of salinity and temperature measurements (9 experiments), without added noise. (**B**) Subset of the time series collected over a day (2 experiments) without added noise. (**C**) The same subset of the time series as in (**B**), but before data processing, we added a noise of 0.05 dB to the measured $S_{11}$ to simulate, for example biofouling effects. An increase in MiSSo-derived *S* coincided with an increase in external noise. The MiSSo time series were obtained using 11 $f_{i}$ frequencies: 2.5166, 2.6124, 2.6170, 2.7355, 2.7401, 2.7454, 2.7918, 2.7940, 2.8427, 2.9331, and 2.9339 GHz.

**Table 1 sensors-22-05915-t001:** Relationship between frequency number and resultant accuracy. The table presents one standard deviation (δT and δS) for *S* and *T* retrievals in the presence of noise (0.05 or 0.1 dB) with a varying number (2, 4, 6, 8, and 11) of used frequencies.

Added Noise:	0.05 dB	0.1 dB
Standard Deviation:	δS/δT [psu/${}^{{}^{\circ}}$C]	δS/δT [psu/${}^{{}^{\circ}}$C]
Number of Frequencies		
2	0.7/0.42	1.2/0.58
4	0.53/0.41	0.87/0.64
6	0.19/0.12	0.38/0.26
8	0.14/0.1	0.26/0.23
11	0.11/0.07	0.24/0.15

**Table 2 sensors-22-05915-t002:** Comparison of selected salinity sensors.

Property	MiSSo	CastAway CTD [18]	SeaBird 41 [7]
Thermistor needed	**NO**	YES	YES
Accuracy [psu]	<0.01	>0.1	>0.1 (thermistor thermal mass limited)
Acquisition time [s]	<0.01	0.2	0.2
Biofouling wipers	YES	NO	NO
Price	LOW	MEDIUM	HIGH

## Data Availability

The data that support the findings of this study are available from the corresponding author, upon request.

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
