# Peer review of "A Salinity–Temperature Sensor Based on Microwave Resonance Reflection"

_sensors, 2022, doi:10.3390/s22155915_

Round 1
Reviewer 1 Report
(1) How are signals [1] and [2] in Fig.1c obtained? It is better to mark them in Fig.1.
(2) Why does an increase in the number of independent frequencies increase the T and S retrieval accuracy? How is the accuracy of 0.01psu or 0.01℃ obtained with 100 consecutive measurements?
Author Response
Reply to the Referee's 1 comment and a list of changes in the revised manuscript
"A salinity-temperature sensor based on microwave resonance reflection"
We appreciate the referee's time, effort, and comprehensive reading of our manuscript and his/her comments that we have addressed below.
The comments from the reviewer are quoted verbatim in italics, followed by our reply and the changes implemented in the revised manuscript. We hope that the referee will find that the changes satisfactorily address his/her concerns.

Reviewer 2 Report
The manuscript is generally well-written and scientifically valid, and its quality is high. However, a number of crucial questions remain to be solved. The manuscript needs to consider these points before being published.
1. 1. The abstract is not very appealing and should be revised.
2. 2. Author should also explain the novelty of this sensor, as the sensor used in your experiment is already on the market and you just integrate two sensors into one.
3. 3. Figure 1b should also show a clear image and information about the different sensors without a dip into the water.
4. 4.The significance of the experiment should be mentioned to make the article more
Logical in the real practical platform with more data.
5. 5. Please show the different days of the image of sensors in continuous mode of working then it will be able to call that it “lowers maintenance costs associated with biofouling”.
6. 6.There should be sufficient explanations for the experimental phenomena in the result section for example “mechanism of the operating sensor”.
7. 7. Author should add a Table which shows an experiment that changes salinity and temperature with respect to the conductivity.
8. 8. Author should also add a comparison table of conductivity/salinity with this integrated instrument and commercially calibrated conductometer at different temperatures for showing accuracy.
Author Response
Reply to the Referee's 2 comments and a list of changes in the revised manuscript "A salinity-temperature sensor based on microwave resonance reflection"
We appreciate the referee's time, effort, and comprehensive reading of our manuscript and his/her comments that we have addressed below. The comments from the reviewer are quoted verbatim in italics, followed by our reply and the changes implemented in the revised manuscript. We hope that the referee will find that the changes satisfactorily address his/her concerns.

Round 2
Reviewer 1 Report
The authors revised the paper according to the reviewers' suggestions. The paper can be accepted in present form.
Author Response
Reply to the Referee’s 1 comment and a list of changes in the revised manuscript
”A salinity-temperature sensor based on microwave resonance reflection” in attached file.

Reviewer 2 Report
Thank you for author response.
Author Response
Reply to the Referee’s 2 comment and a list of changes in the revised manuscript
”A salinity-temperature sensor based on microwave resonance reflection” in the attached file.
